# Acquisition of Competencies of Nurses: Improving the Performance of the Healthcare System

**DOI:** 10.3390/ijerph20054510

**Published:** 2023-03-03

**Authors:** Raquel Ortega-Lapiedra, María Jesús Barrado-Narvión, Jara Bernués-Oliván

**Affiliations:** 1Department of Management, University of Zaragoza, 50009 Zaragoza, Spain; 2IEDIS: Research Institute of Employment, Digital Society and Sustainability, University of Zaragoza, 50018 Zaragoza, Spain; 3Internal Medicine Unit, Royo Villanova Hospital, 50012 Zaragoza, Spain; 4Department of Psychiatry and Nursing, University of Zaragoza, 50009 Zaragoza, Spain; 5Department of Marketing, ESIC Business and Marketing School, 50012 Zaragoza, Spain

**Keywords:** acquisition of competencies, nursing, performance, healthcare system

## Abstract

Perspectives of the core competencies of nurses are varied among postgraduate-year nurses, which makes it challenging to establish training programs and develop evaluation instruments. Particularly critical for nurses is the ongoing acquisition of competencies throughout life. Sometimes this acquisition is funded by the healthcare system, but the key question is how the system leverages this acquisition and ultimately how it translates into patient care. This study seeks to explore nurses’ key competencies acquired through continuing education from the perspective of two groups of postgraduate nurses with different levels of experience and with different objectives to be assessed. An NGT procedure was applied to the group discussion. The participants were recruited according to basic factors such as the number of years of professional experience, their level of education, and their preferred professional status. Thus, seventeen professionals participated in the study, representing two public hospitals in the city. Following the NGT procedure, the competencies identified from the thematic analysis were scored and ranked to achieve a consensus. Eight core issues were derived in the novel group concerning transferring the competencies to patient care quality: holism, care work, organizational barriers, specialization, no transfer, confidence, knowledge, and instrumental tools. Four core issues were derived when asked about the relationship between the resources invested and the organizational and professional development of the nursing staff: professional development, positive learning, negative learning, and recognition. In the more experienced group, seven issues were derived from the first issue raised: continuous learning, quality, confidence, holism, safe care, autonomy, and technical issues. Additionally, six issues arose from the second question: satisfaction, autonomy, creativity, productivity, professional development, and recognition. In conclusion, the perceptions of the two selected groups are negative when it comes to assessing the extent to which the competencies acquired in lifelong learning are transferred to the patient and the system evaluates and recognizes these competencies for improvement.

## 1. Introduction

Healthcare organizations should constantly be searching for ways to improve service quality and its impact on patient care and on the global population in general. Governments seem to devote a large portion of public funds to the training of these professionals, but the question remains: to what extent does this investment in continuous training reach the patient, and does the system allow professionals (nurses) to apply it?

The COVID-19 pandemic has highlighted the shortcomings, in Spain and throughout Europe, in the surveillance field and how difficult it is to manage a health crisis without a robust management model of public health [1]. The need for lifelong learning for the nursing community that allows them to acquire holistic knowledge and skills is increasing in the changing and highly demanding healthcare environment. Nurses have a responsibility to keep up to date on their skills and competencies, but institutions must provide adequate structure and support. The institutional culture must value training as an investment and, in this sense, must promote an environment conducive to learning, which in turn must facilitate the professional development of nurses and, with it, improve their care for patients and for the organization. 

In the period 2020–2022, the activity of nursing professionals increased in complexity and difficulty; together with more demanding patients, this has led to dissatisfaction, stress, and exhaustion. With all certainty, this trend will continue during the coming years. Society must assume responsibility for the importance of healthcare provision to these professionals, which should imply increasing financial resources for the profession and reducing the barriers encountered by nurses, e.g., in aspects related to fewer opportunities for continuous training by their organizations, the high cost of training, loss of motivation, shortage of staff, family issues, and limited involvement of management teams. Consequently, nurses must understand the relationship between the acquisition of competencies through continuous learning and the improvement of patient care and professional practice. 

## 2. Background

### 2.1. Competencies in Nursing

Competence is defined in a variety of ways. Many professions have their core competencies. Nursing competency includes the crucial abilities required for fulfilling one’s role as a nurse. Therefore, it is important to clearly define nursing competency to establish a foundation for the nursing education curriculum. However, while the concepts surrounding nursing competency are important for improving nursing quality, they are still not yet completely developed. In [2], the authors define nursing competencies as: “the habitual and judicious use of communication, knowledge, technical skills, clinical reasoning, emotions, values, and reflection in daily practice for the benefit of the individual and community being served”.

According to a review of the international literature on nursing, competency can follow three theoretical approaches: behaviorism, trait theory, and holism [3]. Behaviorism refers to competency as the ability to perform individual core skills. Trait theory considers competency as the individual traits necessary for effectively performing duties. Holism views competency as a cluster of elements, including knowledge, skills, attitudes, thinking ability, and values required in certain contexts, including professional judgment. We use in this study the holistic view used by [4], who developed the clinical nursing competence self-assessment scale (CNCSS) that considers four competency concepts: (1) basic nursing abilities, (2) the ability to provide care that addresses individual needs, (3) the ability to modify the care environment and collaboration systems, and (4) the ability to devote time toward professional development in nursing practice. The three theoretical approaches complement each other as they go from individual skills to performing duties to taking a step further to a global (holistic) concept in which aspects related to professional development, continuous learning, risk management, or nursing care management are included. It is this last aspect that we have focused on in this study.

In Spain, there is no obligation to take a given number of hours of continuous learning once the nursing professional has obtained their degree. However, in some regions, a climate has been created that favors continuous learning with an accrediting legislative system, supported by the consideration of the time of attendance at training courses organized by the social partners as working time [5].

Because no patients were involved in the study, ethical approval by the hospital ethics committee was not required, but the process of ethical considerations was taken into account with the study participants, just as it is taken into account in other nonhealthcare settings. The investigator explained to the participants the purpose of the research as well as the procedure to be followed on the days of the NGT session. The consent form was distributed to the group of participants because the sessions were recorded, thus protecting their rights.

### 2.2. Continuous Learning in Competencies and Performance

The relationship between continuous learning and results has barely been studied, especially in Spain. Thus, ref. [6] reflects on the importance of lifelong learning and discusses its meaning and its voluntariness or obligatory nature. Although the authors show the lack of reliable data on access to and participation in training courses, the study concludes that nurses have great interest in continuous education and professional development. In this line, the study detects an increase in satisfaction among nurses who participate in lifelong learning, acquiring core competencies and a greater connection to their work, although it does not find a confirmed relationship between lifelong learning and professional development. The research discusses the scarcity of studies that establish the effectiveness of training on the cost-benefit ratio and on patient care once the competencies have been incorporated into daily practice. 

It is clear that the literature that analyzes the competencies acquired through lifelong learning after graduation is inconclusive and scarce when it comes to transferring this to quality and organizational aspects. Previous research shows how the investment in continuing education does not translate into the health system taking advantage of it to improve the quality of care for the patient; it remains an acquisition of competencies that the health professional uses personally without being supported by the health organization, despite the fact that in many cases it is the health system itself that financed this continuing education. 

Ref. [7] analyzes the relationship between the costs and benefits of this continuous learning. The research establishes 33 benefits, of which 18 are personal and 15 are social. Satisfaction is the benefit with the highest score, followed by the pleasure of learning and an increase in new knowledge and techniques, results that are in line with other studies. The study presents evidence of an increase in direct costs, under the nurses’ belief that they must pay direct and indirect costs. Additionally, no cost-benefit ratio is established. 

Ref. [8] investigates, from a qualitative point of view, the perception of nurses about the impact of lifelong learning on patient care and competency acquisition. The study concludes that continuous training is perceived by nurses as a critical issue in providing quality care, but at the same time there is great resistance on the part of management teams to fund that training. 

In the 1990s there was an open debate about the need to determine whether the acquisition of skills through continuous training improves care practice by basing the evaluation of performance on results instead of on the training process itself. Studies that measured the impact on care practice seem to establish a causal relationship between lifelong learning and care. However, [9] concludes that, although the medical institutions showed confidence in the relationships among core competencies, continuous training, and care practice, in more than 35% of studies the measurement instruments are not always valid or reliable, so we must take the conclusions with some caution. 

In light of the nursing literature, it is difficult to conclusively prove a relationship between the quality of patient care and continuous training for competencies acquisition. However, the need for strong associations among certain variables is gradually being incorporated. These variables are, primarily, the individual competence needs of the professional, the context in which they work, the nature of the continuing training, and the maintenance of a professional career. 

In the following decades, the concern with continuous training and its efficacy continued, although much of the fragmented research displays a tendency to evaluate isolated programs without analyzing overall needs and results. On the other hand, the establishment of the obligation, as some groups defend, would not necessarily guarantee the acquisition of certain competencies, nor the improvement of nursing practice; it would simply establish the imperative of measuring results even when it is expensive and difficult, given that patients do not definitively perceive the competences of the nurses in their care practice.

Ref. [7] highlights the substantial literature on impact assessment in terms of university education, although the study notes the scarcity of articles concerning the evaluation of the impacts of continuing education in the health field. In conclusion, lifelong competency training for nurses is a strategic value for the organization and for the professionals. The acquisition of these competencies from a holistic point of view should result in an assessment of the quality of the service as well as the satisfaction of the professional who develops them.

Given that the studies have not established a relationship between continuing education and actual improvement of clinical practice, long-term evaluations must be proposed in a broader way and should include a qualitative perspective.

### 2.3. Aim

Our purpose is to characterize, as a first step, the utility, the impact on patients and institutions, and the influence of the competencies acquired in a holistic sense in the continuous learning after graduation. In the academic field of nursing in Spain, nurses finish their degrees and begin to take a series of courses in a more or less impulsive or reflective way. This is maintained throughout their professional life, with ups and downs determined by work, family, or personal issues. Such courses and seminars are sometimes funded by the national health system, and at other times the nurses are privately funded. This study describes nurses’ experience of working as members of different unit practices in two public hospitals in Aragon (Spain). 

Our study set out to gain a deeper understanding of the lived experience of a group of nurses working in two public hospitals. The global research question is: What is the effect of nurses’ lifelong learning (in terms of competencies acquisition) on healthcare organizations’ performance? The following research questions were proposed to fulfill this goal: Is it possible to transfer competencies acquisition to patient care quality? Is the system capable of harnessing that potential for its own benefit?

The results are used to highlight and offer some practical reflections about public/private investment in lifelong programs and to balance their purposes.

## 3. Materials and Methods

Although there seems to be a constant concern for continuing education to acquire competencies that have an impact on the patient and on the healthcare system in Spain, despite the fact that university training is good, it is not mandatory to take continuing education hours. On an individual basis, sometimes financed by the professional and sometimes by the health system itself, professionals continue to acquire both health and organizational competencies to impact their day-to-day work and professional development. However, there is a climate of dissatisfaction, since training serves only to obtain more points and to consolidate jobs, but in no case does the system evaluate or enable mechanisms to transfer all this intellectual capital to the hospital environment. 

Given this situation, a group of nurses from one of the hospitals analyzed was concerned about analyzing in greater depth the reasons for these aspects.

A descriptive qualitative design was chosen to better understand the subjects of the research questions from the perspective of the nurses themselves (and other process stakeholders). This study adopted a consensus-building approach using the nominal group technique (NGT). The study was undertaken between September and November 2021. 

### 3.1. Sample and Setting

The study settings are two hospitals that both belong to the local government health system inside of the national healthcare system. One of the hospitals is a teaching hospital with more than 800 hospital beds and is located within the city; it is considered the head of the health sector of the region. The other hospital is not a teaching hospital, has more than 250 hospital beds, and collects patients from smaller localities around the city. 

Both hospitals collect the professionals described in the sample. These two hospitals are each a reference in their category of clinical and nonclinical hospitals and were therefore considered a good case study for the research. 

The participants were nurses working in two of the top five hospitals in the region. A purposive sampling strategy was used to ensure that the participants were committed members of the hospital at different stages. Clinical nurse specialists in each hospital provided mailing lists of potential participants, who were invited to participate via an e-mail providing general information on the study’s nature and purpose. Additional nurses were recruited via direct contact during an online meeting and by snowballing. In total, 17 nurses agreed to share their experiences. 

The participants in this study included different levels of training after graduation according to the Spanish education system in nursing professionals: graduates in nursing, postgraduates in nursing with a master’s degree, and postgraduates in nursing with a Ph.D.

Novice group (A): (1) possessed a registered nurse degree; (2) had less than five years of experience as a nurse; (3) half of the sample had a graduate degree and the other half had only a nursing degree.

Experienced group (B): (1) had served as a nurse for more than ten years; (2) one of them had a postgraduate degree; (3) one of them was at the end of their professional life; (4) one of them had served as a head nurse for more than five years.

The participants were 17 nurses working in two public hospitals in the city of Zaragoza, Spain. A purposive sampling strategy was used to ensure that participants were experienced members of the hospital at different stages. 

Regarding the selection of experts, a previous study profile was constructed, consisting of basic factors such as age, gender, level of education, and employment status. In the present study, two groups were established according to their age and professional experience, because nurses work immediately upon finishing their studies (or even before). Group A was formed of 10 nurses (9 female nurses and 1 male nurse) with a maximum age of 40 years. Four of them have postgraduate training (master), one has a doctorate in nursing, four do not have postgraduate training, and one is a specialist nurse. To provide seamless training from basic education to postgraduation clinical practice, many studies have focused on evaluating the nursing competency of university graduate nurses with less than five years of experience. 

Group B was formed of seven nurses (six female nurses and one male nurse) over 40 years old. Only one of them has postgraduate training, with a doctorate. Two are in primary care, one of them being a coordinator. The male nurse is retired so has a complete vision of working life, and the rest are working in the hospital. 

The clinical nurse facilitator (the researcher) provided a mailing list of potential participants, who were invited to participate via e-mail, in which they were given general information on the study’s nature and purpose. 

### 3.2. Data Collection 

The questions revolved around the nurses’ day-to-day experiences in the workplace; supplemental questions were also used to obtain more information. The sample of four discussion groups to four individual interviews seems to be considered adequate for data saturation as the majority of issues across the data were identified. In qualitative research, three to six groups are considered sufficient when the interviews are not stratified by any characteristics of the participants [10]. As the aim of the study was to understand the issues, four individual interviews were carried out to provide richness of the data.

The principal investigator set a date for the NGT sessions and then invited the potential participants via phone or e-mail, which included notification of the inclusion criteria for participation. Only those who fit the criteria were recruited. They were given two weeks to make the decision to participate and sign the consent form. Until they completed the consent forms, they were not divided into groups for the sessions. The rounds took place in a room of the health facility next to one of the selected hospitals. The novice group took about 60–80 min to complete. The most experienced group took about 40–60 min to complete. The rounds were recorded, and the audio files were kept for two years and then deleted and disposed of. 

The questions revolved around the participants seeking solutions to a problem without a record of explicit or structured information, but the solutions are the responsibility of the expert participants or those affected by the problem. Participants’ responses were classified and prioritized in order of importance through consensus. Following the nominal group methodology [11,12], the two questions to be discussed were previously e-mailed so that the “experts” could consider their responses in advance. The questions were as follows:

Has the quality of care for your patients improved in any way after receiving continuous training? This question seeks to answer the research question: Is it possible to transfer the benefits of competency acquisition to patient-care quality? 

How do you think that the time and money invested (direct and indirect costs of training) in the courses taken after finishing “basic” training have influenced your skills and professional development within the public health institution? This question seeks to answer the research question: Is the system harnessing that potential for its own benefit? 

Four sessions were held, and the data were obtained in the order of priority, seeking consensus, since each participant represented a group. Diversity and heterogeneity were sought while avoiding bringing together opposing parties. Subsequently, individual votes generated scores allocated to the acceptance of the various solutions.

### 3.3. Data Analysis

To analyze the data, the qualitative, cyclical, thematic data analysis process described by [11,12] was followed. These authors distinguish between two phases of creative problem-solving: the fact-finding phase and the evaluation phase. 

The present study proposed to use a group diagnostic method with the nominal group technique, as an alternative to obtaining a diagnosis of reliable training needs, as it considers the context of the labor system and includes the opinion and analysis of those involved in the phenomenon being studied.

Each group was individually conducted with one round for each question of the NGT, which process consists of six core tasks: definition, ideas generation, ideas registration, ideas clarification, selection, and prioritization. Step 1: The study purposes, group task, and the NGT procedure were explained to the participants so that they knew the explicit process and understood that each member had an equal opportunity to present their own opinions. Step 2: The group members wrote down the core competencies that they believed to be the most critical for patient care after training (in the case of question 1). Step 3: The members in each group took turns naming one core competency that they had noted, without repeating one that had already been named. After the first round, additional rounds could occur until every member had expressed all of their opinions. During this step, group members had to respect the right of others to speak and could not judge or debate, no matter what idea was being presented. Step 4: Core competencies with similar definitions were combined and simplified through group discussion. Step 5: A vote was taken on each of the core competencies identified in Step 4. An eight-point response from 1 (less important) to 8 (very important) depending on the group and research question was used for each group member to vote for the core competencies. Then, the total score of each core competency was calculated. Finally, in Step 6, each group ranked their core competencies based on the total scores. The group NGT took about 40–60 min to complete.

The steps were repeated in each of the groups (A and B) for the two proposed questions. Group A resulted in eight competencies for question 1 and four issues for the second question. In the case of group B, the first question resulted in seven competencies and the second in six issues. Both cases were categorized in order of importance. 

### 3.4. Ethical Considerations

This study requested ethical approval and was fully approved by the Institutional Review Board in both hospitals. The process of ethical considerations was taken into account with the study participants, just as it is taken into account in other nonhealthcare settings. The principal investigator explained to the participants the purpose of the research as well as the procedure to be followed on the day of the NGT session. The consent form was distributed to the group of participants because the sessions were recorded, thus protecting their rights. The audio files were kept for two years and then deleted and disposed of. 

The regulations governing research in Spain do not require ethical permission for interview studies that do not involve patients, do not cause harm, and do not involve intervention in the physical integrity of the patient. Informed consent was obtained from all respondents before each interview, and participant data were protected by law. This research also includes the following concepts: social value, scientific validity, equitable selection of subjects, and a favorable risk-benefit ratio.

## 4. Results

### 4.1. Nurses’ Experience in Working

Eight subthemes emerged from question 1 and four from question 2 in group A. In group B, there were seven subthemes from question 1 and six from question 2. Table 1 and Table 2 show the prioritized categories for the two group A questions. Table 3 and Table 4 show the prioritized categories for group B. 

#### 4.1.1. Group A. The Novel Group

The nurse preceptor group, responding to question 1, originally proposed eight competencies that lifelong learning had led them to develop. Core competencies were identified and acquired with the improvement of patient care after continuous or permanent training. 

RQ 1: Is it possible to transfer the competencies acquired to patient care quality? 

In a second round, these competencies were prioritized and the group highlighted some institutional problems that negatively affected the patient transfer. Table 1 shows the prioritization of the same: confidence (72 scores), knowledge (68 scores), specialization (49 scores), instrumental tools (48 scores), holism (45 scores), care work (39 scores), organizational barriers (29 scores), and no transfer (15 scores).

RQ 2: Is the system capable of harnessing that potential (continuous learning) for its benefit?

Concerning question 2, which asks for perceptions of the relationship between the investment of resources and professional and organizational development, the panel of experts identified four items that in the second round were prioritized as follows: positive learning (39 scores), negative learning (22 scores), professional development (19 scores), and recognition (19 scores). These items represent the effect on organizational performance (Table 2). 

**Table 1 ijerph-20-04510-t001:** Group A-RQ1: Is it possible to transfer the competencies acquired to patient care quality?

	Nurses’ Scores (10 Responses)										
Holism	5	5	3	6	4	3	4	3	5	7	45
Care work	4	3	5	4	2	4	3	5	4	5	39
Organ. Barriers	1	1	4	7	1	2	8	2	2	1	29
Specialization	3	6	2	5	6	6	2	7	6	6	49
No transfer	2	2	1	2	3	1	1	1	1	2	15
Confidence	8	8	6	7	7	8	5	8	7	8	72
Knowledge	6	7	8	8	8	7	7	6	8	3	68
Instrumental tools	7	4	7	3	5	5	6	4	3	4	48

**Table 2 ijerph-20-04510-t002:** Group A-RQ2: Is the system capable of harnessing that potential (continuous learning) for its benefit?

	Nurses’ Scores (10 Responses)										
Profess. Development	1	1	3	2	2	2	4	2	1	1	19
Positive Learning	4	4	4	4	4	4	3	4	4	4	39
Negative Learning	3	3	1	1	1	2	2	3	3	3	22
Recognition	2	2	2	3	3	1	1	1	2	2	19

#### 4.1.2. Group B. The Experienced Group

RQ 1: Is it possible to transfer the competencies acquired to patient care quality? 

In group B, the first question regarding the improvement of the quality of patient care after training produced seven core competencies identified and acquired for the improvement of patient care after continuous or permanent training. 

Competencies (Table 3): Personal confidence in healthcare practice (38 scores), quality of care (35 scores), holistic vision of the patient (33 scores), expansion of knowledge (31 scores), patient autonomy (28 scores), safe care (25 scores), and technical issues (8 scores). 

**Table 3 ijerph-20-04510-t003:** Group B-RQ1: Is it possible to transfer the competencies acquired to patient care quality?

	Nurses’ Scores (7 Responses)							
Continuous Learning	6	7	6	7	2	1	2	31
Quality	4	5	7	4	5	4	6	39
Confidence	5	6	5	4	4	7	7	29
Holism	7	3	3	3	6	6	5	33
Safe care	3	4	2	5	3	5	3	25
Autonomy	2	2	4	6	7	3	4	28
Technical issues	1	1	1	1	1	2	1	8

RQ 2: Is the system capable of harnessing that potential (continuous learning) for its benefit?

As for the second question in group B (Table 4), the items were as follows: satisfaction (38 scores), autonomy (31 scores), creativity (25 scores), recognition (22 scores), productivity (17 scores), and professional development (14 scores). (The question addresses core competencies identified with the use by the health system or by the working units, in which there was a perceived relationship between the investment of resources and professional and organizational development). 

**Table 4 ijerph-20-04510-t004:** Group B-RQ2: Is the system capable of harnessing that potential (continuous learning) for its benefit?

	Nurses’ Scores (7 Responses)							
Satisfaction	6	6	6	5	3	6	6	38
Autonomy	5	2	5	4	5	5	5	31
Creativity	2	1	4	6	4	4	4	25
Productivity	4	3	3	3	1	1	2	17
Professional Development	1	4	2	1	2	3	1	14
Recognition	3	5	1	2	6	2	3	22

Some important conceptual differences were observed between the two groups. Although both perceived the acquisition of the holistic vision of the patient, in group A, the participants made clear that it came from their academic training, while in group B they considered it a paradigm shift, acquired later with care practice and the skills learned via continuous training. It is also interesting to observe that group A talked in terms of skills, tools, and professional development, while group B spoke in terms of safety, patient autonomy, and humanization of care.

## 5. Discussion

This study, using the NGT method, explores the perspectives of nurses regarding the core competencies acquired with continuous learning, especially their perceptions of how the health system—and specifically its working units—takes advantage of the investment of these professionals in their organizational performance. This analytical study matches a holistic view of competency as a cluster of elements. Possessing professional knowledge and clinical skills is a basic requirement for all healthcare professionals [13,14]. In the most inexperienced group, the competence of safety and confidence as professionals is increased by continuous training, which helps to improve the quality of daily work, and at the same time translates into a continuous motivation to continue learning. The increase in knowledge through training changes the modes of work through daily innovation and making a transition from empirical care to evidence-based care and seeking excellence in treatment. Additionally, it helps the healthcare professionals to specialize by deepening and focusing their skills and knowledge in specific fields. The development of tools for daily practice is an additional contribution that is acquired through continuous training. The perception of the patient from a holistic and systemic perspective is another effect of continuous training, as is empowerment of professionals in their daily care work, while being open to other, less traditional fields related to the nursing profession. The least valued skills are those related to institutional support because, although it generates uneven growth in terms of professional development, the impossibility of applying what they have learned in their daily practice and of developing and applying other skills learned is a drawback, encountering barriers at the institutional level, from senior management, and from colleagues. Continuing education has contributed nothing, in many cases, due to its poor quality, and the fact that it does not expand or supplement academic training, so it is considered solely as an expense. The public health system in Spain is perceived to be of high quality, while those organizations offering online courses are regarded as being of poor quality. There is no degree of consensus as to the quality of the courses. 

In the older group, a reflection was offered from the experience of continuous training throughout their professional life. Problem-solving competence was increased and reflected in daily practice with continuous training. The specialization of care in patients with specific characteristics translated into higher quality care, with more unified and systemic care. The competence of autonomous learning is reinforced when the quality of care is transferred, mainly in the safety and autonomy of the patient to complete the treatment successfully. The reaction to developing more technical or human skills when patient care is considered was much debated, but again there was no consensus, and it was the least valued competence. 

How the institutions or clinical units take advantage of lifelong training is difficult to quantify via a cost-benefit analysis. The nominal technique offers the perceptions of these professionals in a certain environment. In the least experienced group, continuous training serves to improve indicators of optimization of time and resources and encourage research in care as well as developing more efficient care (results). However, it was observed that there is no correlation between the improvement of patient care (results) and professional development because the institution considers only the number of courses taken as a quality indicator to improve its position when choosing certain clinical units, so quality indicators of the use and transfer of that training (results) are not attended to. The perception of the nurses focused more on a transfer at the level of personal development rather than on professional growth.

The most experienced group value the investment of direct and indirect costs in time and money in personal satisfaction, they feel more qualified with better resources, and after years of experience, they are very careful when choosing the training that interests them, not for the consolidation of a job position. However, continuous training has not allowed them to have greater autonomy due to institutional organizational barriers.

## 6. Conclusions

NGT is a quick and effective method for obtaining a consensus of ideas or values. In this study, NGT was used to explore the perceptions of a group of nurses on the core competencies acquired in the continuing education process and on how best to transfer them to patient care and, at the same time, to the medical institution. 

The two groups participating in the study identified several competencies that are acquired through continuing education and that are transferred to patient care depending on the degree of experience of the participating group. Thus, in the less experienced group, eight key competencies were identified: holism, care work, organizational barriers, specialization, nontransfer, confidence, knowledge, and instrumental tools. In the more experienced group, there were seven patient-transferable competencies identified: continuous learning, quality, trust, holism, safe care, autonomy, and technical issues. When asked whether the healthcare system takes advantage of the acquisition of these potential competencies and how it is materialized, group A (newcomers) reflected this in four items: professional development, positive learning, negative learning, and recognition. The group with more years of working life reflected it in six items: satisfaction, autonomy, creativity, productivity, professional development, and recognition. The conclusions of this study can serve as a guideline for preparation courses and can facilitate the development of evaluation instruments or scales for preceptor training.

## Data Availability

Not applicable.

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
