# Peer review of "Acquisition of Competencies of Nurses: Improving the Performance of the Healthcare System"

_ijerph, 2023, doi:10.3390/ijerph20054510_

Round 1
Reviewer 1 Report
Thank you for this relevant interesting paper supporting the future of nursing. This paper reads well and adds a valuable contribution to nursing practice.
Although the introduction reads well in setting the scene for this paper and makes reference to relevant literature it would benefit and be strengthened by further referencing in this area e.g. the beginning is not supported by any references line 28 – 58, line 63 who is this paper by? Lines 74 – 78 for another example could be supported by relevant references.
Regarding the methodology can you clarify if 2 focus groups were used for data collection incorporating the NGT approach?
You do clearly state that no ethical approval was required in your country for this study however, in the section on data analysis you state that the team knew the participants. If I have understood this correctly then can you clarify how you maintained ethical principles - how do you know you did not get proper face data as they knew you and the team?
In the results section - I think it would be better to link directly with the language used in table 1 to the text so reads sharper. The words used in the table do not match the language used in the text.
I am unclear how the tables present the data - using qualitative approach usually involves theme development but you are presenting more of a numbers approach. Were the participants asked to score statements?? More clarity required here
Reviewer 2 Report
Thank you for submitting the manuscript entitled “Acquisition of competences of nurses: Improving the performance of the health care system’ to Nursing under IJERPH. Please see my comments below.
Abstract
· Need to follow the journal instruction for author to prepare the manuscript. The abstract should include a bit background and purpose and the requirement as below.
“We strongly encourage authors to use the following style of structured abstracts, but without headings: (1) Background: Place the question addressed in a broad context and highlight the purpose of the study; (2) Methods: briefly describe the main methods or treatments applied; (3) Results: summarize the article’s main findings; (4) Conclusions: indicate the main conclusions or interpretations. The abstract should be an objective representation of the article and it must not contain results that are not presented and substantiated in the main text and should not exaggerate the main conclusions”
· Suggest to rewrite to allow more understanding for your research report.
Introduction
· Lacking in-text citations. In the 1st paragraph, there is no reference coding. Actually, the whole manuscript should include appropriate and relevant citation to support your statements and arguments.
· Try to avoid using ‘etc’ but you should mention significant items.
· Lines 63 to 74 include abstract thought and ideas. What kind of model for the conceptual analysis you mentioned? How three theoretical approaches link one another and how do they link to your study?
· Line 75 to 78 include unclear message related to the practice in Spain. The authors need to explain how the practice affect their practice.
· Line 81-82, which author? Who is he?
· Lines 90-97, What is the bibliographic approach? Where is it from? What about its content? What research establish benefits?
· What is the meaning of “The study presents evidence of an increase in direct costs, under 95 the nurses belief that they must pay direct and indirect costs. Additionally, no cost-benefit 96 ratio is established.” ? How is it related to the study purpose?
· The paragraph and paragraph are fragmented. The authors should read the relevant papers, digest, and disseminate what they have learned from the evidence. The authors should not write a paragraph for a previous study. It results in loss of direction and too long description.
Sampling and setting
· Purposive sampling was usually used in qualitative studies. How did you recruit the nurse. What were the recruitment criteria? What characteristics of two public hospitals? What only these two hospitals?
· The report on the details of the nurse participants should be in the result section
Data collection
· How about the ethics? Any committee to approve your study?
· Who was the interviewer? Any training to conduct the interview? Who scheduled the interview? Where were the interviews conducted? How long did the interviews take? Was the interviews recorded? What method? Only 4 groups? It’s important to interview until data saturation for qualitative studies.
· Not understand step 5, 8-point response, 1 was less important to 8,4,7. What are 8,4,7?
· According to this section, it seems like a survey including quantitative questions. The authors should revisit the study design and method.
· How did you design the questions? Why did you ask those questions? How were those questions able to address your study purposes? The authors should explain the direction of each item. For example, “satisfaction” what did you want to know? Job satisfaction, workplace environment? Patient care” patient outcome?
· 3.2.2 heading with Group A but the description starting with in group B. Wrong entry?
Results
· Duplicated in the sampling section
· Not clear about the findings.
Discussion: should be rewritten. Since the results were not clear. The result interpretation may not be valid.
Conclusion: Too long
Overall
· The authors need to rewrite if the [n] as the reference paper to be the subject of the sentences.
· The linkage of each idea should reach the study purposes. The description of the methodology must be improved.
· The results are not clear.
· The discussion for result interpretation may be difficult to comprehend.
· Conclusion should be short and focused.
· Lacking in-text citations and inadequate references.
· English proficiency should be checked.
